# Ultrafast light-induced dynamics in the microsolvated biomolecular indole chromophore with water

Jolijn Onvlee [1,2,3], Sebastian Trippel[1,2] & Jochen Küpper [1,2,4] ✉

Interactions between proteins and their solvent environment can be studied in a bottom-up approach using hydrogen-bonded chromophore-solvent clusters. The ultrafast dynamics following UV-light-induced electronic excitation of the chromophores, potential radiation damage, and their dependence on solvation are important open questions. The microsolvation effect is challenging to study due to the inherent mix of the produced gas-phase aggregates. We use the electrostatic deflector to spatially separate different molecular species in combination with pump-probe velocity-map-imaging experiments. We demonstrate that this powerful experimental approach reveals intimate details of the UV-induced dynamics in the near-UV-absorbing prototypical biomolecular indole-water system. We determine the time-dependent appearance of the different reaction products and disentangle the occurring ultrafast processes. This approach ensures that the reactants are well-known and that detailed characteristics of the specific reaction products are accessible – paving the way for the complete chemical-reactivity experiment.

It is a long-held dream of chemistry to follow chemical reactions in real time[1–3]. Here, the observation of the transition state[1] and the recording of electronic and nuclear motion[4,5] during the making and breaking of bonds are of special interest. These processes at the heart of chemistry occur on ultrafast attosecond (as) to picosecond (ps) timescales. One of the fundamental challenges in their investigation is to initiate the reactions effectively instantaneously on the timescale of the dynamics[2,6]. In the ultimate chemical-reaction-dynamics experiment the reactants are well-defined and well-known, the characteristics of the products, e.g., their yields, momenta, and structures, are precisely observed, and the intermediate electronic and nuclear structures are precisely recorded with high spatial and temporal resolution.

In the ongoing quest for this ultimate experiment, scientists designed increasingly advanced machines to prepare well-defined reactants[7,8] and to probe the characteristics of the products in high detail[9]. These ingredients already enabled extremely detailed studies of elementary chemical reactions involving atoms and small molecules[10,11] including full details on the quantum-state-correlations of reactants and products[12,13], albeit without temporally resolving the chemical dynamics.

Alternatively, investigating photochemically-triggered reactions in pump-probe experiments with ultrashort laser pulses enabled the direct study of chemical dynamics in real time[2,14]. Using intense ultrashort laser pulses to excite strong electronic transitions with subsequent photochemical rearrangements and possibly breaking of the molecules has two crucial advantages: it directly defines the starting time of the chemical reaction and it also yields significantly higher densities of molecules undergoing the chemical reaction. These dynamical molecular systems can then be probed using ion-[15,16] or electron-imaging techniques[17,18], high-harmonic-generation spectroscopy[19,20] and laser-induced electron diffraction[4,21,22], or x-ray[23,24] and electron diffraction[3,25].

Severe limitations still exist for such time-resolved studies of bimolecular reaction systems. The ultrafast dynamics of these complex reaction systems are studied through so-called half-collisions[26], in which a chemical bond between two molecules is broken due to the laser excitation. Bimolecular aggregates, so-called molecular clusters,

[1]Center for Free-Electron Laser Science CFEL, Deutsches Elektronen-Synchrotron DESY, Notkestraße 85, 22607 Hamburg, Germany. [2]Center for Ultrafast Imaging, Universität Hamburg, Luruper Chaussee 149, 22761 Hamburg, Germany. [3]Institute for Molecules and Materials, Radboud University, Nijmegen, The Netherlands. [4]Department of Physics, Universität Hamburg, Luruper Chaussee 149, 22761 Hamburg, Germany. ✉e-mail: jochen.kuepper@cfel.de

are produced in a supersonic expansion. However, this yields a broad distribution of cluster sizes and whereas the mean and width of the distribution can be tuned[14,27], it does not allow the production of samples of individual aggregates. In combination with the necessarily broad bandwidth of ultrashort laser pulses, one always simultaneously initiates and probes the dynamics of multiple molecular systems, with different structures and sizes. This leads to overlapping signals from different chemical reactions[14].

Small hydrogen-bonded aggregates of aromatic molecules or chromophores with polar solvent molecules like ammonia ($NH_3$) and water ($H_2O$) are important model systems for the interactions between proteins and their solvent environment[28,29]. For instance, these interactions affect the folding and thereby directly the function of proteins[30]. Microsolvated biomolecular chromophores and their ultrafast dynamics were extensively studied, but current knowledge of their dynamics is strongly limited due to the problem of overlapping signals described above[14,31].

The indole-water (indole($H_2O$), $C_8H_7N \cdot H_2O$) complex, for instance, is of high relevance, as indole is the chromophore of the most strongly near-ultraviolet (UV) absorbing common amino acid tryptophan[32] and thus proteins. It was shown that the environment strongly affects the photochemical properties of tryptophan[33]. These characteristics were frequently used to investigate the structure and dynamics of proteins[34]. In addition, the UV-light-induced dynamics of bare and microsolvated indole ($C_8H_7N$) was studied extensively both theoretically and experimentally[29,31,35–39].

When hydrogen-bonded aggregates of chromophores and polar solvent molecules are irradiated by UV light, they are typically electronically excited to one or several $\pi\pi^*$ states[40], in the case of indole most likely two, i.e., $^1L_b$ and $^1L_a$[41]. These states often possess a conical intersection with an optically dark and dissociative $\pi\sigma^*$ state that plays an important role in the photochemistry of these species[29,42]. For indole in this $\pi\sigma^*$ state it is expected that it ejects an electron into the aqueous environment leading to the formation of a charge-separated state, or solvated electron[32]. For the indole($H_2O$) aggregate it was predicted that this electron-transfer process is followed by the transfer of a proton, leading to a net hydrogen-transfer reaction from indole to $H_2O$[29]. However, so far this hydrogen transfer has not been observed experimentally, although it was observed for the similar indole-ammonia (indole($NH_3$)$_n$) system[43], which was ascribed to ammonia being a better hydrogen acceptor than water[29].

Previous studies on UV-excited indole($H_2O$) clusters found dynamics occurring on multiple timescales of 20...100 fs, 150...500 fs, and 14 ps[31,39]. These were tentatively ascribed to an internal conversion between electronically excited states, relaxation dynamics along the $\pi\sigma^*$ state, and a coupling of the $\pi\sigma^*$ state with the electronic ground state, respectively[31,39]. However, these studies suffered from the overlapping signals of different cluster sizes described above. Therefore, product channels could not be investigated and the long-time relaxation dynamics were dominated by contributions from the fragmentation of the larger clusters.

Here, we present the results of an ultrafast-dynamics pump-probe experiment that utilised the electrostatic deflector to produce a high-purity bimolecular solute-solvent aggregate sample, in combination with velocity-map imaging (VMI) mass spectrometry to disentangle the reaction products. We utilised these capabilities to study the UV-induced dynamics of the prototypical indole($H_2O$) complex including eventual dissociation on the picosecond timescale. We demonstrate that our high-purity sample allows us to investigate product channels and to follow the dynamics of indole($H_2O$) on long timescales. This provides significant additional insight into these prototypical reaction dynamics, i.e., it provides evidence for an incomplete hydrogen-transfer process in indole($H_2O$). Overall, our experimental approach, with the bimolecular reactants well-defined and the product channels clearly identifiable

in real time, propels us a major step forward to unravel the complete pathways in bimolecular reaction systems.

## Results

The ultrafast chemical dynamics of indole($H_2O$) was investigated in a molecular-beam apparatus containing an electrostatic deflector and a velocity-map-imaging ion detector, see Fig. 1c. Using the deflector, we spatially separated indole($H_2O$) from the other species in the molecular beam, e.g., indole, $H_2O$, and helium seed gas[44]. This resulted in a high-purity indole($H_2O$) sample, as shown in Supplementary Note 1. Most signal from a small remaining fraction of ($H_2O$)$_2$ could be discriminated by ion-momentum imaging. UV pump pulses with a central wavelength of 269 nm electronically excited indole($H_2O$). The reaction products were detected through ionisation with a delayed NIR probe pulse with a central wavelength of 1320 nm and velocity-map imaging of the generated ions. The high-purity sample provided by the deflector allowed us to investigate all product channels in an effectively background-free manner, since the product molecules, i.e., indole and water, are not present in the molecular beam in the interaction region—whereas in traditional experiments they were present in the direct molecular beam in large amounts that obscured these product signals. Thus the purified reactants allowed us to identify the appearance of indole$^+$ and $H_2O^+$ product ions that originate from indole($H_2O$).

Figure 1a, b shows the measured indole($H_2O$)$^+$ (blue dots), indole$^+$ (red diamonds), and $H_2O^+$ (green squares) ion signals as a function of the pump-probe delay. The $H_2O^+$ and indole($H_2O$)$^+$ signals are scaled up for improved visibility by factors 10 and 5, respectively. Figure 1a shows the long-time relaxation dynamics, whereas Fig. 1b zooms in on the short delays for indole($H_2O$)$^+$ and indole$^+$. Dynamic effects are clearly visible in all three signals: All channels exhibit a constant ionisation signal for negative delays that increases when the laser pulses temporally overlap. The indole($H_2O$)$^+$ signal shows a fast increase followed by a decay with a fast and a slow component, whereas the indole$^+$ product signal contains a fast increase followed by a slow increase. We could clearly observe a delay of the fast increase in the indole$^+$ signal compared to the fast increase in the indole($H_2O$)$^+$ signal. The $H_2O^+$ product signal shows a slow increase as a function of the pump-probe delay, see Fig. 1a. On top of these general dynamics, we observed oscillations with a period of 1.67 ps for indole($H_2O$)$^+$ and indole$^+$, which are in phase between the indole($H_2O$)$^+$ and indole$^+$ signals. High-temporal-resolution measurements for indole$^+$ at ~120 ps showed that the oscillations are not damped on that timescale, see Fig. S4 in Supplementary Note 5. Our measurements did not reveal any oscillations in the $H_2O^+$ signal.

To disentangle the underlying dynamics of these time-dependent ion yields we employed a reaction model, see Methods. The observations could be described well by a five-level model, allowing to populate the levels sequentially:

$$\text{indole}(H_2O)\, S_0\, (1) \xrightarrow{\text{UV}} \text{indole}(H_2O)\, \pi\pi^*\, (2)$$
$$\xrightarrow{\tau_2} \text{indole}(H_2O)\, \pi\sigma^*\, (3)$$
$$\xrightarrow{\tau_3} \text{indole}(H_2O)\, S_0\, (4) \tag{1}$$
$$\xrightarrow{\tau_4} \text{indole} + H_2O\, (5)$$

with the time constants $\tau_i$ coupling states $i$ and $i+1$. The dynamics in indole($NH_3$)$_n$[43] and indole($H_2O$)$_n$[31] clusters were previously described with similar models, albeit with six and four states. Here, we needed five levels to accurately describe the dynamics we observed in indole($H_2O$). We started with all population in the electronic ground state $S_0$ (1) of indole($H_2O$) and used Maxwell-Bloch equations to find the time-dependent populations of all states. The ion yields are calculated as linear combinations of these populations, which for indole$^+$ and

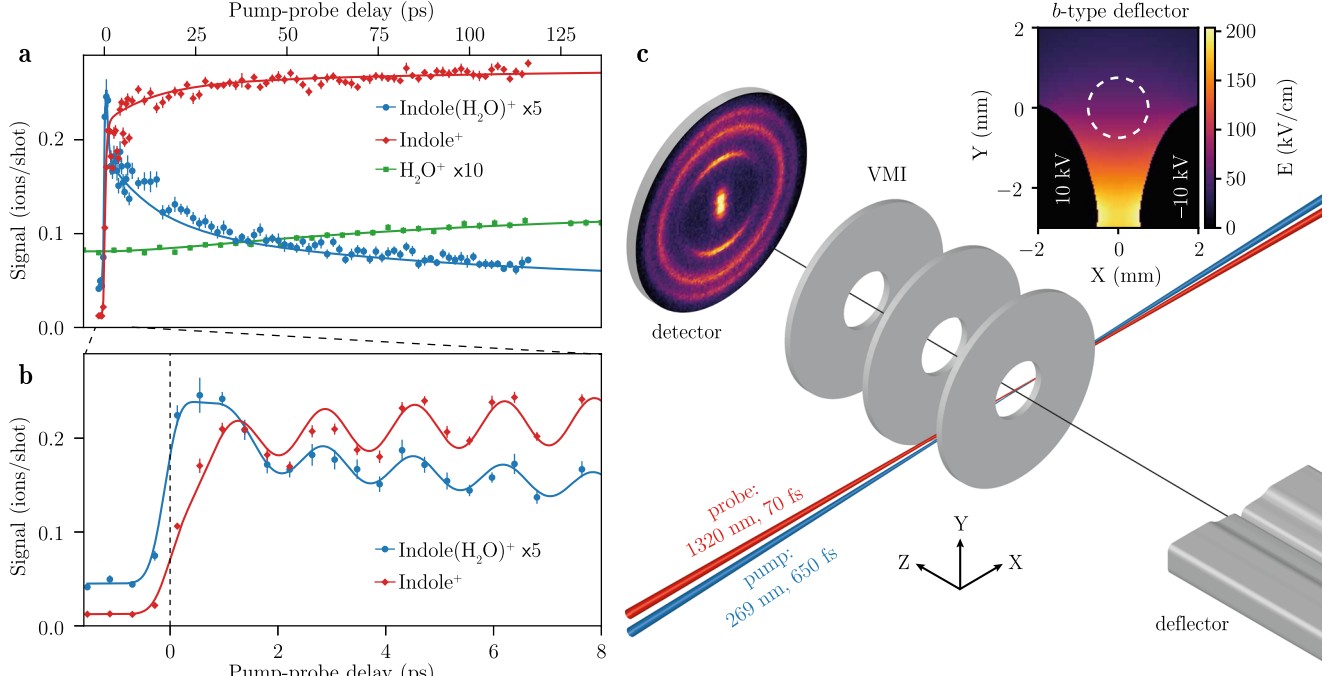

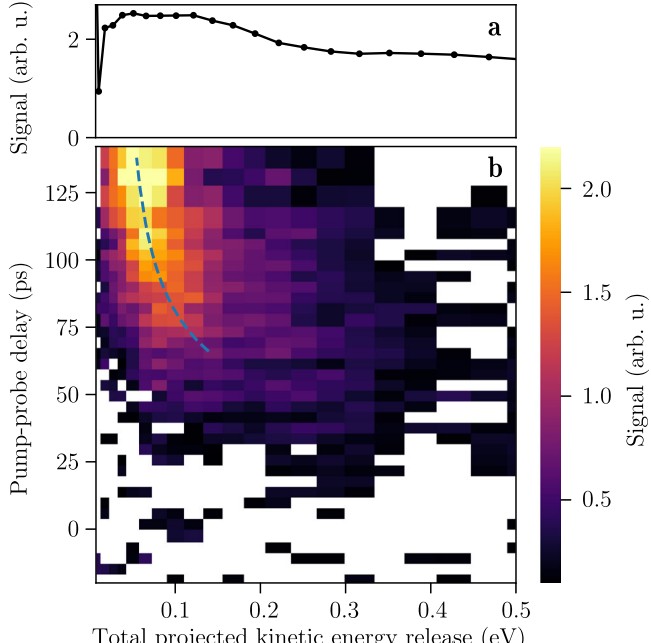

**Fig. 1 | Time-dependent molecular signals and experimental setup. a** Delay-dependent yields of (blue) indole($H_2O$)$^+$, (red) indole$^+$, and (green) $H_2O^+$. Symbols represent the experimental data whereas lines represent the simulated data based on our reaction model. Statistical $1\sigma$ error estimates are given by vertical lines for all experimental data points. **b** Delay-dependent indole($H_2O$)$^+$ and indole$^+$ signals for short delays. **c** Illustration of the experimental setup, containing an electrostatic deflector and a velocity-map imaging (VMI) detector. A pump laser electronically excited indole($H_2O$), and a delayed probe laser ionised the reaction products. See Methods for details. The inset shows a cross section of the deflector: colours indicate the electric field strength $E$ and the white dashed circle indicates where the direct molecular beam passes through the deflector.

---

indole($H_2O$)$^+$ were multiplied by a cosine function representing the oscillations. We note that these timescales, esp. $\tau_2$, do not necessarily correspond to pure electronic interconversion processes, *vide infra* and that the model does not distinguish between the $^1L_b$ and $^1L_a$ electronic states, but rather considers them as one state.

We fitted the coefficients for the linear combinations as well as the time constants to the experimental data and obtained $\tau_2 = 445 \pm 71$ fs, $\tau_3 = 13 \pm 2$ ps, and $\tau_4 = 96 \pm 10$ ps. These results were used to calculate the yields of indole($H_2O$)$^+$, indole$^+$, and $H_2O^+$ shown by the blue, red, and green solid lines in Fig. 1a, b, respectively. Figure 1a shows the signals computed from the Maxwell-Bloch equations directly, whereas Fig. 1b includes the cosine functions describing the oscillations. This model matches the experimental data very well. Additional information about the underlying dynamics was obtained from the projected kinetic-energies of $H_2O^+$, i.e., the delay dependence of the total kinetic energy release (TKER) in the neutral dissociation $C_8H_7N \cdot H_2O \longrightarrow C_8H_7N + H_2O$, see Methods. To separate the delay-dependent signal from the constant $H_2O^+$ signal from dissociation of $(H_2O)_2$, we first determined the mean TKER distribution for negative pump-probe delays, when the $H_2O^+$ ion yield is constant. This distribution, which is shown in Fig. 2a, was subtracted from all TKER resulting in the TKER changes shown in Fig. 2b. For short delays, the TKER distribution is similar to the static background. However, a dynamical signal develops when the delay increases: The mean TKER decreases and the distribution gets narrower.

## Discussion

Based on our measurements and reaction model we expect the dynamics depicted in Fig. 3a to occur. In short: The UV laser excites indole($H_2O$) from the electronic ground state $S_0$ (1) to the electronically excited $\pi\pi^*$ state (2). The system then interconverts to the optically dark $\pi\sigma^*$ state, which is accompanied by nuclear and

**Fig. 2 | Time evolution of the total kinetic-energy release distribution of $H_2O^+$. a** Total kinetic-energy release (TKER) distribution for negative pump-probe delays, based on the projected kinetic-energy distribution of $H_2O^+$. **b** Time-dependent TKER-distribution differences from the neutral dissociation of indole($H_2O$) after subtraction of the delay-independent signal. The blue dashed line represents the mean TKER obtained from an ion-dipole-interaction model, see Methods.

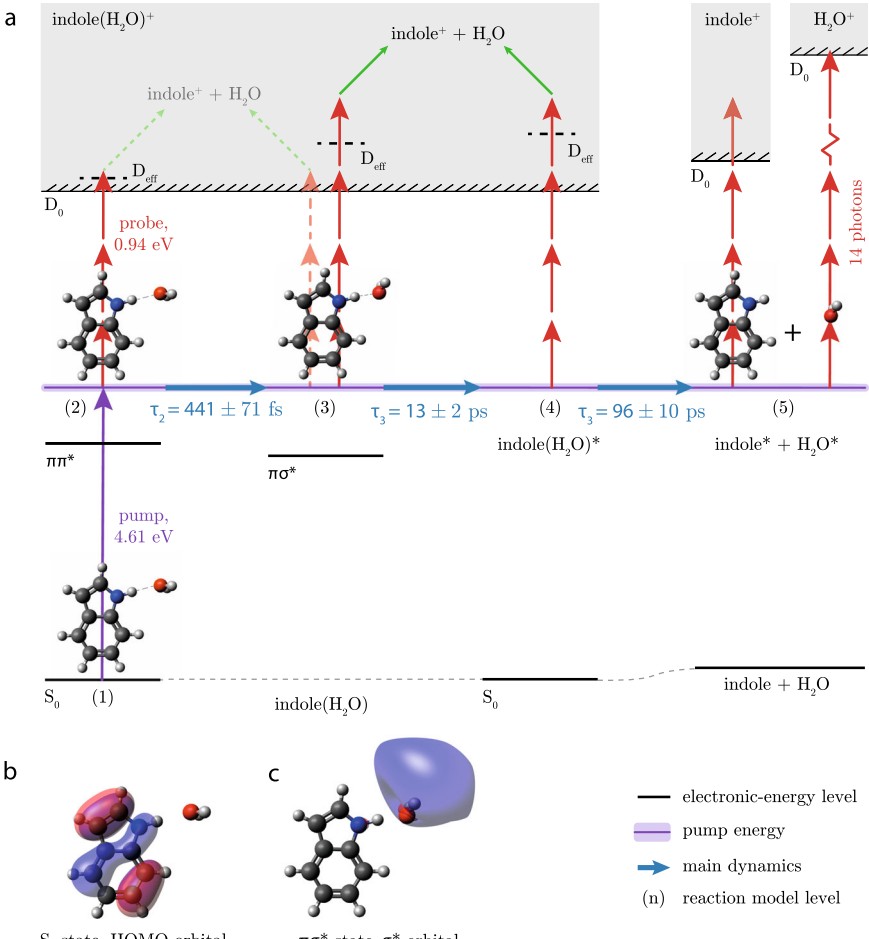

**Fig. 3 | Schematic of the UV-induced photochemistry in indole(H₂O). a** The purple arrow indicates the UV pump-photon energy yielding the overall energy in the system after absorption of an UV photon, indicated by the purple line, which is conserved throughout the dynamics. The shaded purple area represents the corresponding bandwidth. Blue horizontal arrows depict the different steps in the observed photochemistry with corresponding timescales; see text for details. The red arrows indicate the absorption of NIR probe photons leading to ionisation. The black dashed lines labelled $D_{eff}$ represent the cationic states that need to be accessed to efficiently ionise from a particular initial state in indole(H₂O); see text

for details. The green arrows represent the dissociative ionisation process, leading to the formation of indole⁺ and H₂O. Dashed arrows indicate a significantly lower probability for a specific process to take place. The indole + H₂O product channel contains a lot of vibrational energy. Note that the energies in this figure are not to scale. (**b**) and (**c**) Computed isosurface representations of the relevant molecular orbitals of indole(H₂O). **b** The highest occupied molecular orbital (HOMO), a $\pi$-orbital, for the equilibrium geometry in the $S_0$ state. **c** The $\sigma^*$ orbital in the equilibrium geometry of the $\pi\sigma^*$ state.

electronic rearrangements into (3). Subsequently, interconversion takes place to the vibrationally hot $S_0$ state (4). This vibrationally-hot electronic-ground-state indole(H₂O) dissociates into separate vibrationally-excited indole and H₂O molecules (5), most likely *via* so-called statistical unimolecular decay. Moreover, vibronic-wavepacket dynamics occur in this system, which were taken into account in the reaction model. In the following, the different steps will be discussed in more detail.

When the UV laser (purple arrow in Fig. 3a) excites indole(H₂O) to the $\pi\pi^*$ state, the system can be ionised by the subsequent absorption of three NIR photons (left red arrows) through the electronic ground state $D_0$ of the cation, which has a similar geometry as the $\pi\pi^*$ state[45] and an ionisation energy of $E_i$=7.37 eV[46]. The vertical transition is indicated by the dashed horizontal line labelled $D_{eff}$ in Fig. 3a. This results in the fast initial increase in the indole(H₂O)⁺ signal. Due to the excess energy above the ionisation potential, fragmentation of indole(H₂O)⁺ can occur[44], leading to a small contribution to the fast increase in the indole⁺ signal, see Supplementary Note 4.

The subsequent interconversion from the $\pi\pi^*$ state to the $\pi\sigma^*$ state with $\tau_2 = 445 \pm 71$ fs is accompanied by nuclear rearrangement

and electron transfer, as shown by the results of ab initio calculations that are described in Methods. In the equilibrium geometry of the $\pi\sigma^*$ state, depicted in Fig. 3c, the H₂O molecule tilts along the in-plane-bending coordinate of the cluster and the N-O distance decreases, whilst the N-H distance increases with respect to the geometry in the $S_0$ state, depicted in Fig. 3b. Moreover, an electron is transferred from a $\pi$-orbital on the indole moiety to a $\sigma^*$-orbital localised around the H₂O molecule. This is depicted by the calculated isosurfaces of the orbital amplitudes plotted in the structures in Fig. 3b, c.

In order to efficiently ionise the system from this geometrically different $\pi\sigma^*$ state with a different amount of vibrational energy (3), i.e., in a vertical transition, it seems that a higher-energy cationic state needs to be accessed, depicted by the dashed horizontal line labelled $D_{eff}$ in Fig. 3a. We suspect that this cationic state is above the binding energy of indole(H₂O)⁺ of 0.6 eV[47], resulting in more fragmentation of the cation and correspondingly more indole⁺ signal[47]. This process likely required four NIR photons, see Supplementary Note 4 for more details. Overall, this leads to a decrease in the indole(H₂O)⁺ signal and a corresponding increase in the indole⁺ signal. This agrees with the fast increase in the indole⁺ signal being delayed with respect to the fast increase in the indole(H₂O)⁺ signal.

We attribute the 1.67 ps oscillations overlayed on these electronic and nuclear dynamics to wavepacket dynamics, for instance due to indole ring breathing modes or C−H-stretch vibrations, which modified the transition strengths in the system, see Supplementary Note 5. We expect that these dynamics occurred due to the coherent excitation of vibrational states with an energy difference of 20 cm$^{-1}$, within the bandwidth of the pump laser depicted by the shaded purple area in Fig. 3a, and that they manifest themselves as a modification of the vertical-ionisation probability, which lead to the observation of these oscillations.

Subsequently, the indole($H_2O$) complexes in the $\pi\sigma^*$ state interconvert within $\tau_3 = 13 \pm 2$ ps to the $S_0$ state through nonadiabatic coupling, as was described before[48]. We expect that this interconversion leads to vibrationally-excited indole($H_2O$) clusters in the electronic ground state, which were ionised less likely due to the increased $E_i$ for a vertical transition−depicted by the dashed horizontal line $D_{eff}$ state in Fig. 3a – and which results in indole($H_2O$)$^+$ ions that subsequently dissociate, leading to indole$^+$ signal. This would explain the slow decrease in the indole($H_2O$)$^+$ signal and the slow increase in the indole$^+$ signal.

We believe that the vibrationally-hot indole($H_2O$) subsequently dissociates on a timescale of $\tau_4 = 96 \pm 10$ ps into separate neutral indole and $H_2O$ molecules containing significant vibrational energy. Both products could be ionised, which explains the increasing indole$^+$ and $H_2O^+$ signals. At least four NIR photons are needed to reach the ionisation energy for the vibrationally-hot indole, whereas a minimum of 14 NIR photons is needed for the ionisation of ground-state $H_2O$.

An alternative pathway for the formation of $H_2O$ molecules would be direct dissociation in the $\pi\sigma^*$ state. However, our measured TKER distributions, Fig. 2, match Maxwell-Boltzmann distributions for delays >65 ps, which strongly indicates that the $H_2O$ molecules are formed in a statistical unimolecular-decay process in the $S_0$ state instead of a direct dissociation process in the $\pi\sigma^*$ state, see Methods. Direct dissociation from the $\pi\pi^*$ state is in principle possible but, also because of the short sub-picosecond lifetime of this state, irrelevant.

The mean of the TKER distributions changes as a function of the delay, see Fig. 2. This time evolution is explained by considering the ion-dipole interaction: When the $H_2O$ molecule is ionised and the separate indole and $H_2O^+$ molecules are still relatively close, they will repel each other due to the repulsion of the $H_2O^+$ charge and the indole dipole, leading to an increase in the kinetic energy. For longer delays, the increased distance between the two moieties results in a correspondingly lower kinetic energy. The width of the TKER distribution is governed by the wavepacket evolution along the ion-dipole interaction potential and decreases for longer delays. We used a classical ion-dipole interaction model to describe the evolution of the TKER; details are provided in Methods. This yielded the dashed blue line in Fig. 2, which is in very good agreement with our experimental results, indicating that the two moieties, on average, slowly move apart with a speed $v \approx 12$ m/s. This is another confirmation for the statistical unimolecular-decay process in the $S_0$ state.

In order to shed light on the possible net hydrogen-transfer process taking place in the $\pi\sigma^*$ state of indole($H_2O$)[29], we investigated the $H_3O^+$ signal. However, we did not find any delay-dependent effect in the $H_3O^+$ signal coming from neutral dissociation of indole($H_2O$). This indicates that, although the N−H bond is stretched in the $\pi\sigma^*$ state, the electron transfer is not followed by proton transfer to $H_2O$ for dissociation. We conclude that after UV excitation at 4.61 eV the indole($H_2O$) aggregate either survives or dissociates into the individual indole and $H_2O$ molecules, which themselves stay intact and slowly move away from each other.

Our pump-probe experiments on pure indole($H_2O$) samples in combination with our reaction model based on a five-level system provide new insight into the ultrafast processes occurring in this prototypical solvated-biomolecule system, especially regarding the combined electronic and nuclear dissociation dynamics and its product channels. The time constant $\tau_2 = 445 \pm 71$ fs that we assigned to the interconversion from the $\pi\pi^*$ to the $\pi\sigma^*$ state and accompanied nuclear rearrangement and electron transfer is within the broad range $\tau_2 = 150 \ldots 500$ fs previously tentatively attributed to dynamics on the $\pi\sigma^*$ surface[39]. We note that $\tau_2$ in the present study is on the same order as the instrument response function (IRF). We could not resolve the fastest time constant of ~ 50 fs[31] or 20...100 fs[39] observed before, since it is shorter than the IRF. This time constant was previously assigned to internal conversion from the $\pi\pi^*$ to the $\pi\sigma^*$ state[31] or from the $^1L_a$ to the $^1L_b$ state[39], which are both $\pi\pi^*$ states. Regarding the time constant $\tau_3 = 13 \pm 2$ ps, our interpretation is consistent with previous tentative assignments of a 14 ps decay constant to interconversion from the $\pi\sigma^*$ state to the $S_0$ state[31]. While previous work was not able to distinguish products from different cluster sizes, our well-defined-reactant study fully supports this finding. Furthermore, the $\tau_4 = 96 \pm 10$ ps time constant we obtained for the bond breaking and formation of $H_2O$ is comparable to the ones found for the production of $(NH_3)_{n-1}NH_4^+$ resulting from dissociation of indole($NH_3$)$_n$ clusters with $n \le 2$[43].

In conclusion, we demonstrated that our experimental approach using species-selected samples to perform pump-probe studies of ultrafast chemical dynamics provides unprecedented details on intermediates and reaction products and thus the chemical reaction dynamics. Creating a high-purity sample of the prototypical solvated biomolecule indole($H_2O$) enabled us to investigate its UV-induced dissociation dynamics in intimate detail, well beyond previous experimental studies. We observed an initial delay in the appearance of indole$^+$ ions, which we ascribed to the ionisation of indole($H_2O$) from the $\pi\pi^*$ and $\pi\sigma^*$ states resulting in different cationic states of indole($H_2O$) with distinct fragmentation probabilities. Moreover, we could follow the long-time relaxation dynamics in the reaction products, which revealed clear evidence for an incomplete hydrogen-transfer process and thus indicates that the indole chromophore is protected by the attached water against UV-induced radiation damage. This is opposite to earlier theoretical predictions[29], but fully in line with previous experiments[31].

While in previous experiments the $H_2O^+$ product signal was completely obscured by the unavoidable large amount of $H_2O$ in the molecular beam, our purified samples allowed us, for the first time, to experimentally determine the time constant of the hydrogen-bond-breaking process to $\tau_4 = 96 \pm 10$ ps. Moreover, based on the kinetic-energy distributions of the $H_2O$ products we conclude that this biochemically important process[49] occurs *via* statistical unimolecular decay in the electronic ground state. As ultrafast excited-electronic-state deactivation after the absorption of UV photons could be essential for the photostability of proteins[50], our results demonstrate how such mitigation of UV-induced radiation damage through solvent interactions works.

Overall, these results demonstrate that our experimental approach combining the deflector, velocity-map imaging, and pump-probe ultrafast time-resolved spectroscopy enables the observation of complete chemical-reactivity pathways in chemical reactions of complex molecular systems. The investigation of a bimolecular half-collision reaction allowed for the precise triggering of the dynamics and presents a promising approach for more complex chemical systems.

Our approach can directly be combined with shorter, i.e., few-femtosecond or attosecond, laser pulses and with tunable wavelengths to follow the energy-dependent ultrafast electronic and chemical processes in complex reaction systems. This could be further aided by coincidence measurements[16,51]. On the other hand, high-energy UV photons could be used in order to ionise the complex and its fragments with a single photon. Moreover, diffractive imaging of the nuclear dynamics[4,24,25] would provide complementary detailed information on the actual atomic structural dynamics. For instance,

photoelectron-momentum-imaging and laser-induced-electron-diffraction experiments would be a direct extension of the current experiments and initial experiments are ongoing[52,53]. These approaches should reveal the complete reaction pathway of the nuclear and electronic dynamics that occur in these reactions. Such detailed insights will ultimately yield a deep understanding of the formation and breaking of bonds and allow to develop a truly dynamical basis of chemistry.

## Methods

### Experimental setup
The experimental setup[54,55] is shown schematically in Fig. 1c. 95 bar of helium was bubbled through room-temperature water before passing through the sample reservoir of an Even-Lavie valve containing indole (Sigma-Aldrich, $\geq 99\%$). The valve was operated at 110 °C and a repetition rate of 250 Hz. After passing through two skimmers, the beam travelled through a 15.4 cm long electrostatic deflector[8]. Applying a voltage of 20 kV between the electrodes of the deflector created inhomogeneous electric fields that spatially dispersed and separated indole and indole($H_2O$) based on their Stark effect[8,44], see Supplementary Note 1. Passing through another skimmer the molecules were intersected by the focused pump and probe laser beams in the centre of a VMI spectrometer.

Indole($H_2O$) was electronically excited by ultraviolet-light (UV) pulses with a central wavelength of 269 nm, a pulse duration of ~650 fs (full width at half maximum, FWHM), and a peak intensity of ~$2 \cdot 10^9$ W/cm². Near-infrared (NIR) pulses centred around 1320 nm with a duration of ~70 fs (FWHM) and a peak intensity of ~$1 \cdot 10^{13}$ W/cm² were used to ionise the complex and its fragments. We used a wavelength of 1320 nm to avoid the three-photon resonant excitation of indole and indole($H_2O$) at 800 nm and to have a similar ionisation step as in previous work on UV-induced dynamics in indole($H_2O$)$_n$ clusters[39]. Both laser beams were linearly polarised along Y, i.e., parallel to the detector plane, and focused into the centre of the molecular beam with $4\sigma \approx 90$ μm and $4\sigma \approx 60$ μm for the intensities of the UV and NIR beams, respectively. Unless mentioned otherwise, we adjusted the laser powers such that the ionisation signal of indole($H_2O$) with either of the two beams alone was negligible.

The generated ions were accelerated towards a multichannel-plate and phosphor-screen detector using the VMI spectrometer. Images were recorded with a CMOS camera at 500 Hz, alternating between molecular-beam signal and background frames. The MCP was temporally gated in order to record ion images for individual mass-to-charge ratios, which we refer to as mass-gated images. Indole$^+$ and indole($H_2O$)$^+$ signals were measured in a detuned-VMI mode to avoid saturation and damage of the central part of the detector. $H_2O^+$ ions were measured in VMI mode at an increased NIR intensity of ~$1 \cdot 10^{14}$ W/cm², accounting for the relatively high ionisation energy $E_i \approx 12.6$ eV of $H_2O$.

The delay between pump and probe laser pulses was scanned back and forth multiple times using a motorised translation stage. Indole($H_2O$)$^+$ and indole$^+$ signals were recorded in the same measurements, by scanning the pump-probe delay and the mass-gating simultaneously. We used a step size of 417 fs for $t = -1.535 \ldots 6.805$ ps, made one step of 834 fs, and used a step size of 1.668 ps for $t > 7.639$ ps. We used ~8000 laser shots per data point. The $H_2O^+$ signal was recorded for $t = -5.705 \ldots 140.245$ ps with a step size of 4.17 ps and ~41,000 laser shots per data point. Data at 119.395, 123.565, and 140.245 ps was discarded due to instabilities in the spatial overlap of the UV and NIR beams for the longest delays. The delays $t > 114$ ps do not significantly alter the fit, but show that the $H_2O^+$ signal starts to level off. To improve the statistics for Fig. 2, this data was combined with a second measurement for which the $H_2O^+$ signal was recorded for $t = -19.391 \ldots 122.390$ ps with a step size of 1.668 ps and ~340,000 laser

shots per data point. The two data sets were merged and averaged over 4 ps, and the data points at 118, 122, and 142 ps were discarded due to the non-perfect spatial overlap of the laser beams.

### Reaction model
We described the time-dependent ion signals using a reaction model for a five-level system, see Eq. (1). The Maxwell-Bloch equations that describe the evolution of the populations $\rho_{ii}$ of state $i$ corresponding to this model are given by[14,48]

$$
\begin{aligned}
\dot{\rho}_{11} &= \frac{i}{2}\Omega_0 g(t)(\rho_{12} - \rho_{21}) \\
\dot{\rho}_{22} &= -\frac{i}{2}\Omega_0 g(t)(\rho_{12} - \rho_{21}) - \Gamma_{22}\rho_{22} \\
\dot{\rho}_{21} &= -\frac{i}{2}\Omega_0 g(t)(\rho_{11} - \rho_{22}) - (\Gamma_{21} - i\Delta\omega)\rho_{21} \\
\dot{\rho}_{12} &= \frac{i}{2}\Omega_0 g(t)^*(\rho_{11} - \rho_{22}) - (\Gamma_{21} + i\Delta\omega)\rho_{12} \\
\dot{\rho}_{33} &= \Gamma_{22}\rho_{22} - \Gamma_{33}\rho_{33} \\
\dot{\rho}_{44} &= \Gamma_{33}\rho_{33} - \Gamma_{44}\rho_{44} \\
\dot{\rho}_{55} &= \Gamma_{44}\rho_{44},
\end{aligned}
\tag{2}
$$

where $\Gamma_{ii} = 1/\tau_i$ and $g(t) = \exp(-\frac{1}{2}(t/\tau_{IRF})^2)$, which represents the instrument-response function (IRF) with $\tau_{IRF} = 381$ fs, see Supplementary Note 3. We assumed $\Delta\omega = 0$ and $\Gamma_{21} = \Gamma_{22}/2$[14]. The Rabi frequency was estimated to $\Omega_0 = 3.4$ ps$^{-1}$ based on the peak intensity and duration of the UV pulses and an estimated transition dipole moment $\mu_{12} \approx 15\ e$ pm[48].

Initially, all population was in state 1, i.e., $\rho_{11}(-\infty) = 1$. Integrating Eq. (2) yields the delay-dependent populations of the different states. The simulated ion-signal intensities for indole($H_2O$)$^+$, indole$^+$, and $H_2O^+$ are given by linear combinations of the different populations $\rho_{ii}$:

$$
\begin{aligned}
I_{\text{indole}(H_2O)^+}(t) &= p^{osc}_{\text{indole}(H_2O)^+}(t) \sum_{i=2}^{4} A_i \rho'_{ii}(t) \\
I_{\text{indole}^+}(t) &= p^{osc}_{\text{indole}^+}(t) \sum_{i=2}^{5} B_i \rho'_{ii}(t) \\
I_{H_2O^+} &= C\rho'_{55},
\end{aligned}
\tag{3}
$$

with the population $\rho'_{ii}(t) = f(t) \otimes \rho_{ii}(t)$ of state $i$ after convolution with a Gaussian function $f(t)$ with a FWHM of 70 fs, which represents the intensity envelope of the NIR pulse[14]. The decay constants $\tau_2$, $\tau_3$, and $\tau_4$ as well as the coefficients $A_i$, $B_i$ and $C$ in Eq. (3) were fit using a Levenberg-Marquardt algorithm and a reduced-$\chi^2$ objective function of simulated against background-corrected experimental ion signals, Fig. 1a. The oscillations in the indole$^+$ and indole($H_2O$)$^+$ signals were modelled as $p^{osc}_j(t) = a_j + b_j \cos(\omega t + \phi)$, with $j = $ indole($H_2O$)$^+$ or indole$^+$. These parameters were optimised once using a high-resolution measurement of the oscillations and then fixed in the fitting procedure to $a_{\text{indole}(H_2O)^+} = 1.04$, $b_{\text{indole}(H_2O)^+} = 0.14$, $a_{\text{indole}^+} = 0.92$, $b_{\text{indole}^+} = 0.11$, $\omega = 2\pi \cdot 0.60$ THz $= 3.77$ THz and $\phi = 1.77$.

The best fit yielded $\tau_2 = 445 \pm 71$ fs, $\tau_3 = 13 \pm 2$ ps, and $\tau_4 = 96 \pm 10$ ps with a reduced $\chi^2$ of $\chi^2_\nu = 1.42$ and a coefficient of determination $R^2 = 0.999$. The resulting time-dependent contributions of the individual states to the ion signals are shown in Fig. S3 in Supplementary Note 4.

**Evolution of the total kinetic energy release.** For the low-kinetic-energy $H_2O^+$ signal resulting from neutral dissociation, we investigated the kinetic-energy distributions as a function of the pump-probe delay. The total projected kinetic energy release (TKER) assuming neutral

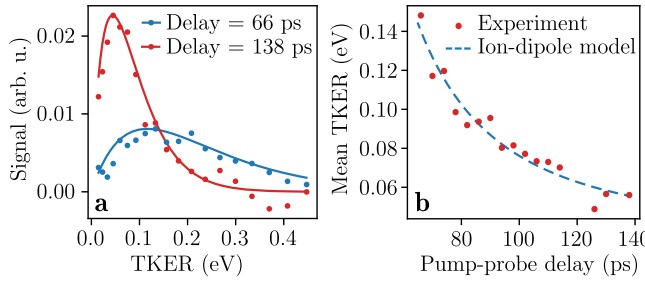

**Fig. 4 | Evolution of the total projected kinetic energy release. a** TKER distributions for pump-probe delays of 66 ps (blue) and 138 ps (red). The dots are the experimental distributions after subtracting the mean TKER distribution for negative pump-probe delays, whereas the solid lines show fitted statistical distributions. **b** Mean TKER values obtained from the experimental distributions (red dots) and a fit through these data points (blue dashed line) based on an ion-dipole interaction model, see the text for details.

dissociation can be computed using

$$\text{TKER} = E^K_{H_2O} \frac{m_{\text{indole}(H_2O)}}{m_{\text{indole}}}, \tag{4}$$

where $E^K_{H_2O}$ is the kinetic energy (KE) of $H_2O$ and $m_{\text{indole}}$ and $m_{\text{indole}(H_2O)}$ are the masses of indole and indole($H_2O$), respectively.

Figure 2 shows how the projected TKER distributions evolve as a function of the pump-probe delay. The blue and red dots in Fig. 4a represent these distributions for delays of 66 ps and 138 ps, respectively. We subtracted the mean TKER distribution for negative pump-probe delays, shown in Fig. 2a, which represents the delay-independent $H_2O^+$ signal. We note that the recorded projected KEs are, in principle, different from the real three-dimensional KEs, but in line with the geometric orientation due to the dipole excitation and the clear rings in the VMIs we assume the projected KEs are a reasonable approximation for the following discussion.

We successfully fitted Maxwell-Boltzmann distributions to all experimental velocity distributions for delays >65 ps. The resulting experimental KE spectra and statistical distributions for delays of 66 ps and 138 ps are shown in Fig. 4a. This indicates that the $H_2O$ molecules resulted from statistical unimolecular decay, in the vibrationally-hot electronic ground state, instead of direct dissociation from an electronically excited state, e.g., the πσ* state[36].

From the data in Fig. 2 it is clear that the mean TKER decreases, i.e., the mean kinetic energy of the $H_2O$ molecules, decreases as a function of the delay. The distribution also becomes narrower for longer delays, in line with a statistical process in which the hottest reactants dissociate the fastest and earliest. To visualise the temporal evolution of the TKER, the mean values of the fitted statistical distributions as a function of the pump-probe delay are shown in Fig. 4b. This temporal evolution of the TKER distribution could be explained by an ion-dipole interaction model, detailed below, based on previous descriptions of similar Coulomb-repulsion effects, e.g., in the dissociation dynamics of small stable molecules[56–58].

When a vibrationally-hot indole($H_2O$) molecule dissociates and is afterwards ionised by the probe laser an electron can in first instance tunnel back and forth between the indole and $H_2O$ moieties. However, when the distance between the two molecules increases, the electron at some point localises on one of the two monomers. If the hole localises on $H_2O$ we get indole + $H_2O^+$ and can detect the $H_2O^+$ ion. Since the indole side pointing towards the $H_2O^+$ ion is the positive end of the indole dipole moment in the $S_0$ state[59], the two moieties will repel each other due to the ion-dipole interaction. The interaction energy $V$ between indole and $H_2O^+$, neglecting higher-order terms such

as the ion-induced dipole and dipole-dipole interactions, is given by

$$V(t) = \frac{q_{\text{ion}} \mu \cos\theta}{4\pi\epsilon_0 R(t)^2}, \tag{5}$$

with the charge $q_{\text{ion}}$ of the $H_2O$ molecule, i.e., $q_{\text{ion}} = 1$, the dipole moment μ of the indole molecule, the interaction angle $\theta$, the vacuum permittivity $\epsilon_0$, and the distance $R$ between the ion and the dipole. We assumed that μ = 1.96 D, i.e., the dipole moment of indole in the $S_0, v = 0$ state[59], and that $\theta$ is time-independent.

We further assumed that the two fragments dissociate with a constant velocity $v$ starting from an equilibrium distance $R_{\text{eq}}$ at a delay $t_d$. We used the following expression for the time-dependent ion-dipole distance $R(t)$

$$R(t) = R_{\text{eq}} + v(t - t_d), \tag{6}$$

which we substituted in Eq. (5). The total kinetic energy in the system is now given by:

$$\text{TKER}(t) = E_{\text{pump}} + V(t) - E_a, \tag{7}$$

with $E_{\text{pump}}$ the energy of a pump-laser photon, 4.61 eV, and $E_a$ the asymptotic internal energy of the two fragments, i.e., for $R = \infty$.

Fitting $R_{\text{eq}}$, $v$, $t_d$, $\theta$ and $E_a$ to the experimental evolution of the mean TKER yielded the blue dashed line in Fig. 2 and Fig. 4b. We found that $R_{\text{eq}} = 490$ pm, $v = 12$ m/s, $t_d = 51$ ps, $\theta = 38°$, and $E_a = 4.57$ eV. The fit reproduces the experiment well, which indicates that the ion-dipole interaction drives the observed evolution of the TKER. The low asymptotic TKER, i.e., $E_{\text{pump}} - E_a$, which is below 50 meV, is an additional indication for the $H_2O$ molecules resulting from dissociation in the $S_0$ state instead of the repulsive πσ* state, since such repulsive states generally lead to a higher asymptotic energy.

We note that our simple model neglects attractive interactions such as the ion-induced dipole interaction, which could possibly be relevant due to the relatively high polarisability of indole. Moreover, μ could change significantly due to vibrational excitation. However, our experimental data does not contain sufficient information to accurately take this into account and corresponding literature values were not available. As these effects partly cancel each other and based on the very good agreement of our model and experimental data, our results still provide clear insight into the origin of the TKER of $H_2O^+$ signal and, correspondingly, the actual dissociation dynamics of indole($H_2O$).

## Ab initio calculations

We performed ab initio calculations on indole($H_2O$) using PSI4[60]. The ground state geometry was optimised using density-fitted second-order Møller-Plesset perturbation theory (DF-MP2) calculations using an aug-cc-pVTZ basis set. We used equation-of-motion coupled-cluster singles and doubles (EOM-CCSD) in combination with an aug-cc-pVDZ basis set to optimise the geometry in the πσ* state.

## Data availability

The data that support the findings of this study are available from the repository at https://doi.org/10.5281/zenodo.7024411.

## Code availability

The script used to solve the Maxwelll−Bloch equations is available from the repository at https://doi.org/10.5281/zenodo.7024411.

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

## Acknowledgements

We thank Joss Wiese, Melby Johny, and Oriol Vendrell for fruitful discussions and Jovana Petrovic and Terry Mullins for support of the experiments. We acknowledge financial support by Deutsches Elektronen-Synchrotron DESY, a member of the Helmholtz Association (HGF), also for the provision of experimental facilities and for the use of the Maxwell computational resources operated at DESY. This work has been supported by the Clusters of Excellence "Centre for Ultrafast Imaging" (CUI, EXC 1074, ID 194651731) and "Advanced Imaging of Matter" (AIM, EXC 2056, ID 390715994) of the Deutsche Forschungsgemeinschaft (DFG) and by the European Research Council under the European Union's Seventh Framework Programme (FP7/2007-2013) through the Consolidator Grant COMOTION (614507). J.O. gratefully acknowledges a fellowship by the Alexander von Humboldt Foundation.

## Author contributions

J.K. conceived the experiments and supervised the study. J.O. performed the experiment with contributions from S.T. J.O. analysed the data. All authors were involved in interpreting the data and discussing the results. J.O. and J.K. wrote the manuscript with contributions from S.T.

## Funding

## Competing interests

The authors declare no competing interests.
