## [Peer Review File · Nature Communications]

REVIEWER COMMENTS

Reviewer #1 (Remarks to the Author):

This manuscript reports ultrafast time-resolved pump-probe studies of the decay of indole-H₂O clusters following photoexcitation at ~269 nm, with strong field ionisation (SFI, by near infrared (NIR) photons at 1320 nm) plus velocity map imaging of the resulting parent and fragment ions as the probe. A noteworthy feature of the study is the use of an electrostatic deflector to enable spatial separation of the indole-H₂O clusters from the more abundant (non-complexed) indole and water molecules in the molecular beam source. The time resolved ion signals are fit to three time constants (for which the authors provide a rationalisation based on a plausible five state model) plus a cosine function to mimic oscillations observed in the indole⁺ and indole-H₂O⁺ ion signals. These are all positives, and the data and interpretation do offer refinements of the current picture of the non-radiative decay of systems such as this involving a heteroaromatic molecule complexed to (one or more) H₂O, NH₃, etc. molecules.

But I also have some questions about the manuscript. The authors will surely have thought about the work more carefully than most reviewers might have time to do, so these comments should be regarded as questions rather than criticisms. I would be happy for the authors to act on them as they see fit.

The authors emphasise use of a high-purity bimolecular solute-solvent aggregate sample (early on p. 2) and a high purity indole(H₂O) sample (p. 3). Figure S1 shows successful discrimination between indole⁺ and indole(H₂O)⁺ ions with the deflection field on. But the authors will be aware of the extensive literature demonstrating the delicacy of larger neutral and ionic clusters. The expansion conditions are quite severe. Are the authors confident that their beam does not contain a significant fraction of indole-(H₂O)_{n>1} clusters, which are not revealed upon SFI because the larger cluster ions fragment?

Is there a contradiction on p. 3, where the authors first highlight how their technique allows investigation of all product channels in a background-free manner, but in the next column explain how they need to separate the delay-dependent signal from the constant H₂O⁺ rest-gas background signal?

I have no issue that the best-fit value (and uncertainty) of τ_2 is first reported to a precision of 1 fs but question the appropriateness of carrying that over-precision into the subsequent discussion, given that the UV pulse duration is ~650 fs (main manuscript p. 7) or even 692 fs (FWHM) as reported in the Supplementary Information (SI).

I am puzzled by the relative magnitudes of the early time indole-H₂O⁺ and indole⁺ signals. Have I missed an explanation as to why the latter is ~5-times larger in fig. 1? The main text (p. 4) recognises possible fragmentation of indole-H₂O⁺ clusters but suggests that this makes only a small contribution to the indole⁺ yield.

The authors (reasonably to my mind) interpret the 'immediate' appearance of indole-H₂O⁺ signal to $\pi^* \leftarrow \pi$ excitation of the aromatic chromophore within the indole(H₂O) complex, followed by SFI of the resulting $\pi\pi^*$ species by the NIR probe pulse. Guided by prior ultrafast time-resolved studies and speculative conclusions derived therefrom, the authors attribute the kinetics of these excited indole(H₂O) species in terms of (fast, $\tau_2 \sim 450$ fs) coupling from the photoexcited $\pi\pi^*$ state to the $\pi\sigma^*$ potential that promotes N-H bond extension and eventual sampling of the $\pi\sigma^*/S_0$ conical intersection (with a time constant $\tau_3 \sim 13$ ps). The authors propose (p. 4) that efficient ionisation from the $\pi\sigma^*$ state requires promotion to a 'higher energy state of the cation', that lies above the binding energy of the indole-H₂O⁺ cation. It was not clear to me how much of this discussion was based on (intelligent) speculation and how much was 'solid' and based on prior literature: exciting bare indole with a broad band pulse centred at ~ 269 nm would likely sample both the 1La and 1Lb absorption bands (two different $\pi^* \leftarrow \pi$ transitions) – I have not checked to see how these bands shift when the indole is complexed to an H₂O. The $\pi\sigma^*$ state is probably best viewed as having significant (3s) Rydberg character in the vertical Franck–Condon region, with the electron density developing σ^* valence character (localised on the N–H bond) as R(N–H) increases. In general, one might expect a higher ionisation probability from a Rydberg orbital to its corresponding cation limit than from a nearby π^* valence orbital – but this expectation appears to run counter to the arguments advanced in the paper.

The authors suggest (p. 4) that the 1.67 ps oscillations – which endure for >100 ps – are attributable to wavepacket dynamics in the C–H stretch vibrations, that modify the transition strengths in the system and give rise to the observed beating. Whilst not doubting the attribution to wavepacket dynamics, nor that the dynamics are associated with spectator modes in the indole chromophore, the C–H vibrations must be about the most 'passive' modes in this photoexcitation process; the pump and probe steps are both much more likely to promote various ring breathing motions, for example.

Reviewer #3 (Remarks to the Author):

This manuscript describes a femtosecond time-resolved ion-yield measurement on an indole-water beam created by electrostatic deflection. The indole-water complex is excited by a 650-fs long ultraviolet pulse and its dynamics are probed by strong-field ionization with a 1320-nm laser pulse. The authors interpret the time-resolved ion yields in terms of a five-level kinetic model, which consists of photoexcitation to the $\pi\text{-}\pi^*$ state, followed by internal conversion to the $\pi\text{-}\sigma^*$ state, electronic relaxation to the S_0 ground state and statistical dissociation on the ground state. In addition to the ion yields, the authors also measured and interpreted the kinetic-energy release (KER) of the H₂O⁺ ions generated by the probe pulse.

The novelty of this work resides in the fact that excited-state dynamics are studied in a species prepared with the electrostatic-deflection technique. This is a very welcome and timely

development, which opens many interesting possibilities. For this reason, I believe that this manuscript could eventually be published in Nature Communications.

However, although the experimental results are of high quality and the presentation of the data is compelling, I have some reservations with regards to the interpretation of the experimental data. I would suggest that the authors support their interpretation with more rigorous theoretical results, including dynamical calculations, which are nowadays well within reach using standard quantum-chemistry packages.

Specifically, my recommendations are the following:

- 1) It is not clear how to reconcile the potential-energy surfaces shown in Fig. 3 with the ionization and dissociation energies given in the figure and the text. The horizontal line at 7.37 eV and the repulsive curve dissociating to indole+ + H₂O are both supposed to represent the electronic ground state of indole-H₂O⁺. I suggest that the authors calculate the actual potential energy curve of this electronic ground state and display it in Fig. 3.
- 2) As drawn, Fig. 3 suggests that 1UV + 3IR photons can reach 7.76 eV, but the sum of the photon energies is only 7.43 eV. Probably 4-photon processes are not much less likely than 3-photon processes at 10¹³ W/cm² and the authors actually observe 4-photon ionization?
- 3) The authors assign tau₂=445 fs to the pi-pi*  pi-sigma* electronic relaxation. This is unusually slow for this type of internal conversion. Moreover, this interpretation contrasts with Ref. 31, which assigned a ~50-fs time scale to this process. Although Ref. 31 did not resolve the individual indole-water clusters, it is hard to imagine a factor 10 difference of this time scale as a function of cluster size. I suggest that the authors verify their interpretation by comparison with dynamical calculations.
- 4) Related to 3), it would be helpful to understand why the UV pulse was so long (650 fs), in spite of the 4.5 nm bandwidth. Was it stretched on purpose? If it was stretched, it must be highly chirped and the chirp might have an influence on the excited-state dynamics.
- 5) The experimental work is of high quality and the study of excited-state dynamics in species separated by electrostatic deflection is innovative. I nevertheless believe that the title "solvated biomolecules" is an exaggeration and should be toned down to describe more accurately what is being reported.
- 6) The explanation that is given for the oscillation with a period of 1.67 ps is not clear. The authors argue that the 20 cm⁻¹ interval corresponds to the difference of the frequencies of C-H stretching vibrations. I am not convinced by this explanation because the absence of dephasing suggests that only two quantum states are prepared and it is hard to imagine such a selectivity among the C-H stretching vibrations of indole. I suggest that the authors carefully study other possible explanations for this observation.

Overall, I view the reported experimental work as a very important and promising development at the forefront of ultrafast molecular science. Unfortunately, I find the current interpretation is not sufficiently solid or convincing to recommend publication at this point.

Title: Ultrafast light-induced dynamics in solvated biomolecules: The indole chromophores with water

Authors: Jolijn Onlvee, Sebastian Trippel, and Jochen Küpper

This manuscript presents first data from a novel instrument in which electric deflection is used to interrogate the indole-H₂O complex with femtosecond pump-probe pulses free from interference from either monomer or larger clusters. The time-dependent signals due to the intact complex, the indole⁺ fragment, and H₂O⁺ are recorded over a range of delay times between the actinic UV pump pulse and a 1.36 μm multiphoton ionization pulse.

There is no doubt that this work is unique in combining electric deflection, fs pump-probe, and mass-selective velocity-map ion imaging. In this sense, there may be some leeway for the method having within it the possibility for misinterpretation. However, since indole-H₂O has been studied in such detail in the past, the authors are intent on providing a detailed account of every bit of the time-dependent signal in all channels.

I am a little unclear as to what Nature Communications allows for length of manuscripts, but this one is far longer than I would anticipate for a communication. And the SI material is very substantial, and in fact almost essential for making sense of the main text.

1. So many of my questions arise from the use of a 1320 nm photoionization laser. The authors should state explicitly why they chose this multiphoton process for the ionization step. It would have been much cleaner to have a single photon step with a near-UV photon.
2. How far above the S₁ origin are you exciting with the pump laser? Timescales for photophysical processes, dissociation are sensitively dependent on this excess energy. Are you able to dissociate on the S₁ surface?
3. Figure 3: The asymptote to indole + H₂O in the grd state is far lower than shown. This must be the H₂O dissociation asymptote shown associated with the πσ* state? The authors should put both asymptotes on their diagram.
4. Ascribing the 1.67 ps oscillations in the ion signal due to beating between slightly different frequency CH stretch vibrations (something that is only clear from reading the SI material carefully), seems quite unlikely to this reviewer. The language in the main text has to better clarify that they are ascribing this to a difference in frequencies of vibrations. C-H stretches almost certainly wouldn't be responsible for modulating the photoionization cross section, and also should have no Franck-Condon activity in the S₀-S₁ transition nor in the S₁-ion transition. The fact that the quantum beat is a single frequency is beautiful, but also not unique to this study, as the authors point out in the SI. Since indole monomer has the same periodicity quantum beat, I would suggest the authors look at the FC active vibrations in indole monomer, and see whether there are aromatic ring distortion modes that could be involved.

5. The kinetic scheme presented in equation (1) gives the false impression that the four processes occur in parallel; however, in the explanation, the authors clearly are assuming that these are sequential processes. Please clarify this in the text. For instance, how is indole⁺ produced within 1 ps if processes 1-4 are sequential?
6. The authors assume in their analysis that all indole⁺ signal comes from complexes that have dissociated after UV excitation. The alternative is that some of the indole⁺ signal arises from photofragmentation of the ionized complex. It seems unlikely to me that experimental conditions stopped absorption of the 1.36 μm laser pulse after a 3-photon process to form indole-H₂O⁺, while in indole⁺ the process involves at least 4 photons. Did the authors determine any power dependence to their ion signals?
7. The weak time dependence to the H₂O⁺ signal – are you fitting with a zero offset of 0.08 ions/shot?
8. At times, it is hard to filter out what in their explanations comes from previous work and what they are deducing on their own. For instance, on p. 4 of the manuscript (upper right), the authors state that “the N-O distance decreases, whilst the N-H distance increases with respect to the geometry in the S₀ state.” However, their experiment has no way of deducing this.

A couple of minor points:

1. The authors should state explicitly whether there was any orientation dependence in the VMI images.
2. The authors use sometimes misleading language, seemingly in order to make a connection to other issues. Two examples: (1) ‘solvated electron’ when in fact this same orbital shape is present in the absence of the H₂O molecule in the indole monomer. (2) ‘bimolecular solute-solvent aggregate’ rather than ‘indole-H₂O complex’.

Response to reviewer comments

Reviewer 1

This manuscript reports ultrafast time-resolved pump-probe studies of the decay of indole-H₂O clusters following photoexcitation at ~269 nm, with strong field ionisation (SFI, by near infrared (NIR) photons at 1320 nm) plus velocity map imaging of the resulting parent and fragment ions as the probe. A noteworthy feature of the study is the use of an electrostatic deflector to enable spatial separation of the indole-H₂O clusters from the more abundant (non-complexed) indole and water molecules in the molecular beam source. The time resolved ion signals are fit to three time constants (for which the authors provide a rationalisation based on a plausible five state model) plus a cosine function to mimic oscillations observed in the indole⁺ and indole-H₂O⁺ ion signals. These are all positives, and the data and interpretation do offer refinements of the current picture of the non-radiative decay of systems such as this involving a heteroaromatic molecule complexed to (one or more) H₂O, NH₃, etc. molecules.

Thank you very much for the positive evaluation of our work.

But I also have some questions about the manuscript. The authors will surely have thought about the work more carefully than most reviewers might have time to do, so these comments should be regarded as questions rather than criticisms. I would be happy for the authors to act on them as they see fit.

We also thank the reviewer for her/his valuable questions and input.

The authors emphasise use of a high-purity bimolecular solute-solvent aggregate sample (early on p. 2) and a high purity indole(H₂O) sample (p. 3). Figure S1 shows successful discrimination between indole⁺ and indole(H₂O)⁺ ions with the deflection field on. But the authors will be aware of the extensive literature demonstrating the delicacy of larger neutral and ionic clusters. The expansion conditions are quite severe. Are the authors confident that their beam does not contain a significant fraction of indole-(H₂O)_{n>1} clusters, which are not revealed upon SFI because the larger cluster ions fragment?

We are fully confident that our beam does not contain a significant fraction of indole(H₂O)_{n>1} clusters.

We have carefully considered this question for a decade, starting from our first demonstration of the separation of indole(H₂O) (Trippel *et al.*, 2012), and we are convinced that all larger clusters indole(H₂O)_{n>1} are deflected less than the indole(H₂O)_{n=1} system. In addition, we would have seen these larger clusters, if present, using our strong-field-ionisation ion-imaging scheme, as we did in a previous experiment (Trippel *et al.*, 2018). If we would have had a significant amount of larger clusters in our beam, we would have seen their signatures in the deflection profile – at smaller deflections – for indole(H₂O), as we observed before (Trippel *et al.*, 2018).

Is there a contradiction on p. 3, where the authors first highlight how their technique allows investigation of all product channels in a background-free manner, but in the next column explain how they need to separate the delay-dependent signal from the constant H₂O⁺ rest-gas background signal?

We realise that we did not phrase this sufficiently careful and specific in the manuscript and we have improved the updated manuscript correspondingly.

With “background-free” we referred to the fact that there are no time-dependent background signals, in addition to very little static background in the investigated reactant and product molecules, e. g., some remaining water “rest-gas background” is unavoidable in any vacuum/molecular beam experiment.

However, in “traditional” molecular-cluster experiments, the indole⁺ and H₂O⁺ signals would

be heavily dominated by signals from indole and H_2O being present in large amounts *in* the molecular beam in the interaction region, as we specified in the manuscript (l. 150–157). These large signals from the monomers as well as additional signals from larger clusters, which would show dynamic signals themselves, could overwhelm any signals from the indole(H_2O) dimers and thus strongly obscure the product signals we are interested in. Furthermore, as pointed out by the reviewer above, clusters can fragment and thus no clear correspondence between the observed ion masses and the original neutral species is possible.

Now, in our experiments using the deflection-purified samples, for any practical purpose neither the monomer molecules and their strong signals nor larger clusters are present in the interaction region and therefore do not lead to disturbing background signals.

As shown in the Supplementary Information, we do have a constant H_2O^+ background signal coming from dissociation of $(\text{H}_2\text{O})_2$, since $(\text{H}_2\text{O})_2$ is deflecting further than indole(H_2O). As explained in the Supplementary Information, we could clearly identify the origin of this constant delay-independent H_2O^+ signal such that we could confidently separate the signals from each other.

To clarify this, we provided more details and explanation about this, esp. in the *Results* section of the manuscript (l. 143–145)

Most signal from a small remaining fraction of $(\text{H}_2\text{O})_2$ could be discriminated by ion-momentum imaging.

and (l. 219–223)

To separate the delay-dependent signal from the constant H_2O^+ signal from dissociation of $(\text{H}_2\text{O})_2$, we first determined the mean TKER distribution for negative pump-probe delays, when the H_2O^+ ion yield is constant.

I have no issue that the best-fit value (and uncertainty) of τ_2 is first reported to a precision of 1 fs but question the appropriateness of carrying that over-precision into the subsequent discussion, given that the UV pulse duration is ~ 650 fs (main manuscript p. 7) or even 692 fs (FWHM) as reported in the Supplementary Information (SI).

We agree with the general impression of the reviewer and, therefore, always specified both the value and the error estimate of these numbers at any point they are given in the manuscript.

We believe that printing a rounded value, which the reviewer comment implies, would only lead to wrong quotations and use of these rounded numbers. On the other hand, specifying the error alongside every specification of the fitted value keeps the reader fully informed and choose themselves which exact value to use – for interpretation or further work.

I am puzzled by the relative magnitudes of the early time indole- H_2O^+ and indole $^+$ signals. Have I missed an explanation as to why the latter is ~ 5 -times larger in fig. 1? The main text (p. 4) recognises possible fragmentation of indole- H_2O^+ clusters but suggests that this makes only a small contribution to the indole $^+$ yield.

We explained these relative magnitudes of the early time indole(H_2O) $^+$ and indole $^+$ signals on page 4 of the manuscript. Ionisation from the $\pi\pi^*$ state indeed leads to a small amount of fragmentation (l. 254–259). However, ionisation from the $\pi\sigma^*$ state leads to a large amount of fragmentation (l. 279–287).

As shown by Mons *et al.* (1999), [46 in the manuscript]), the fragmentation ratio increases rapidly when the total sum of photon energies is significantly larger than the ionisation energy E_i of indole(H_2O) and indole. Therefore, we expect that a higher-energy cationic state needs to be accessed for ionisation from the $\pi\sigma^*$ state, as detailed in the Methods section (previously in the SI): Figure 4 (previously Figure S4) also shows the contribution of ionisation from the two different states to the indole $^+$ signal. Here, it can clearly be observed that the main contribution arises from ionisation from the $\pi\sigma^*$ state.

The authors (reasonably to my mind) interpret the ‘immediate’ appearance of indole-H₂O+ signal to $\pi^* \leftarrow \pi$ excitation of the aromatic chromophore within the indole(H₂O) complex, followed by SFI of the resulting $\pi\pi^*$ species by the NIR probe pulse. Guided by prior ultrafast time-resolved studies and speculative conclusions derived therefrom, the authors attribute the kinetics of these excited indole(H₂O) species in terms of (fast, $\tau_2 \sim 450$ fs) coupling from the photoexcited $\pi\pi^*$ state to the $\pi\sigma^*$ potential that promotes N-H bond extension and eventual sampling of the $\pi\sigma^*/S_0$ conical intersection (with a time constant $\tau_3 \sim 13$ ps). The authors propose (p. 4) that efficient ionisation from the $\pi\sigma^*$ state requires promotion to a ‘higher energy state of the cation’, that lies above the binding energy of the indole-H₂O+ cation. It was not clear to me how much of this discussion was based on (intelligent) speculation and how much was ‘solid’ and based on prior literature: exciting bare indole with a broad band pulse centred at ~ 269 nm would likely sample both the 1La and 1Lb absorption bands (two different $\pi^* \leftarrow \pi$ transitions) – I have not checked to see how these bands shift when the indole is complexed to an H₂O. The $\pi\sigma^*$ state is probably best viewed as having significant (3s) Rydberg character in the vertical Franck–Condon region, with the electron density developing σ^* valence character (localised on the N–H bond) as R(N-H) increases. In general, one might expect a higher ionisation probability from a Rydberg orbital to its corresponding cation limit than from a nearby π^* valence orbital – but this expectation appears to run counter to the arguments advanced in the paper.

We agree with the referee that we should distinguish better between speculation and knowledge based on prior literature. We updated the manuscript accordingly. Please see the PDF file with changes marked for the details.

We indeed sample both the L_b and L_a states with the the broad band pump pulse used in our experiments. Both states slightly shift in energy when a water molecule is attached to indole (Sobolewski and Domcke, 2000, [32 in the manuscript]).

As shown Figure 3c of the manuscript, resulting from *ab initio* calculations we performed for indole(H₂O), the $\pi\sigma^*$ state indeed has significant Rydberg character. This was also shown in previous theoretical papers (Sobolewski and Domcke, 2000, [32]), where they also write "When a single water molecule is present, the σ^* electron cloud already detaches from the NH group and attaches to the water molecule. The π orbital, on the other hand, is little affected by the complexation with water."

In previous studies on indole(H₂O) (Lippert, 2004, [47 in the manuscript]), femtosecond photoelectron-photoion coincidence (FEICO) measurements using a pump wavelength of 250 nm and a probe with a wavelength of 400 nm showed that the contribution of the two-probe-photon signal compared to the one-probe-photon signal increased within the first ps. This was attributed to the change in electronic structure due to the internal conversion from the $\pi\pi^*$ to the $\pi\sigma^*$ state. Although the ionisation probability to ionise from a Rydberg orbital might be higher, as the referee suggests, it seems that the energy needed to reach the corresponding cation limit, now denoted by D_{eff} in the manuscript and Figure 3, increases as well.

The authors suggest (p. 4) that the 1.67 ps oscillations – which endure for >100 ps - are attributable to wavepacket dynamics in the C-H stretch vibrations, that modify the transition strengths in the system and give rise to the observed beating. Whilst not doubting the attribution to wavepacket dynamics, nor that the dynamics are associated with spectator modes in the indole chromophore, the C-H vibrations must be about the most ‘passive’ modes in this photoexcitation process; the pump and probe steps are both much more likely to promote various ring breathing motions, for example.

Unfortunately, our experiments do not provide information on the vibrations that lead to the oscillations we observe. We speculated that the C-H stretch vibrations could lead to these oscillations, but did not want to exclude other possibilities. We agree with the referee that ring breathing motions could certainly also lead to such oscillations. We updated the wording in the manuscript, see the PDF file with changes marked, and included, in the supplementary

information, some examples of ring breathing motions that also differ by $\sim 20 \text{ cm}^{-1}$ and could, therefore, also lead to the oscillations.

Reviewer 2

This manuscript presents first data from a novel instrument in which electric deflection is used to interrogate the indole-H₂O complex with femtosecond pump-probe pulses free from interference from either monomer or larger clusters. The time-dependent signals due to the intact complex, the indole⁺ fragment, and H₂O⁺ are recorded over a range of delay times between the actinic UV pump pulse and a 1.36 μm multiphoton ionization pulse. There is no doubt that this work is unique in combining electric deflection, fs pump-probe, and mass-selective velocity-map ion imaging. In this sense, there may be some leeway for the method having within it the possibility for misinterpretation. However, since indole-H₂O has been studied in such detail in the past, the authors are intent on providing a detailed account of every bit of the time-dependent signal in all channels.

Thank you for the positive words. In fact, we indeed feel that the rich and detailed quantitative data we could acquire on this seminal and highly important model system using our novel experimental approach deserves a detailed analysis and presentation to the community.

I am a little unclear as to what Nature Communications allows for length of manuscripts, but this one is far longer than I would anticipate for a communication. And the SI material is very substantial, and in fact almost essential for making sense of the main text.

We believe that our manuscript is within the length specifications of *Nature Communications* and, importantly, that the length is fully appropriate for the scientific content. To help the reader, we moved the SI section "Evolution of the total kinetic energy release" to the Methods section, such that one does not have to go back and forth that often between the main manuscript and the SI. We checked that the length of the Methods section still complies with the length specifications of *Nature Communications*.

1. So many of my questions arise from the use of a 1320 nm photoionization laser. The authors should state explicitly why they chose this multiphoton process for the ionization step. It would have been much cleaner to have a single photon step with a near-UV photon.

We believe that our strong-field/multi-photon-ionisation approach using non-resonant near-infrared (NIR) photons provided a more generic ionisation scheme than near-UV photons, as suggested by this reviewer. The latter would yield significantly stronger dependencies on near-resonant ionisation transitions and potentially even be too low in energy to ionise some of the products, especially so for the water products. In our approach, a "simple" change of pulse energy, i. e., intensity, was sufficient from the beginning to adjust for such effects.

We also point out that there was a previous study of this system with a very similar ionisation wavelength of 1305 nm (Peralta Conde *et al.*, 2012, [39 in the manuscript]). We added this in the Methods section (l. 480–484):

We used a wavelength of 1320 nm to avoid the three-photon resonant excitation of indole and indole(H₂O) at 800 nm and to have a similar ionisation step as in previous work on UV-induced dynamics in indole(H₂O)_n clusters (Peralta Conde *et al.*, 2012, [39]).

On the other hand, a deep-UV photon at 10–20 eV would surely be a clean single-photon and very general ionisation source, but the related efforts would compose another huge effort and were beyond our capabilities and the scope of the present study. We, therefore, added a sentence to the last paragraph of the discussion section (l. 440–442):

On the other hand, high-energy UV photons could be used in order to ionise the complex and its fragments with a single photon.

2. How far above the S_1 origin are you exciting with the pump laser? Timescales for photophysical processes, dissociation are sensitively dependent on this excess energy. Are you able to dissociate on the S_1 surface?

The S_1 origin transition of indole(H_2O) is at $\sim 35100\text{ cm}^{-1}$ (Kortner *et al.*, 1998, [37 in the manuscript]). Therefore, we provide approximately 2353 cm^{-1} of rovibrational energy to the system. Assuming a binding energy for the S_1 state similar to the binding energy of the neutral ground state of $\sim 1700\text{ cm}^{-1}$ (Mons *et al.*, 1999, [46]), the molecule could indeed fragment in the S_1 state. However, at this low energy the process will be even slower and less likely than in the highly vibrationally excited S_0 state and practically all molecules will undergo the ultrafast electronic dynamics instead of this direct dissociation from S_1 .

Therefore, and in line with previous work, we believe that the contributions of dissociation from S_1 is fully negligible. Nevertheless, updated the manuscript to point this out (l. 331–333):

Direct dissociation from the $\pi\pi^*$ state is in principle possible but, also because of the short sub-picosecond lifetime of this state, irrelevant.

3. Figure 3: The asymptote to indole + H_2O in the grd state is far lower than shown. This must be the H_3O dissociation asymptote shown associated with the $\pi\sigma^*$ state? The authors should put both asymptotes on their diagram.

We improved Figure 3 based on other comments and suggestions, and put the asymptote for indole + H_2O lower. We refrained from putting the asymptote for H_3O dissociation in as well, since we did not see any H_3O^+ signals and, therefore, do not have any indications for H-atom transfer to occur.

4. Ascribing the 1.67 ps oscillations in the ion signal due to beating between slightly different frequency CH stretch vibrations something that is only clear from reading the SI material carefully), seems quite unlikely to this reviewer. The language in the main text has to better clarify that they are ascribing this to a difference in frequencies of vibrations. C-H stretches almost certainly wouldn't be responsible for modulating the photoionization cross section, and also should have no Franck-Condon activity in the S_0 - S_1 transition nor in the S_1 -ion transition. The fact that the quantum beat is a single frequency is beautiful, but also not unique to this study, as the authors point out in the SI. Since indole monomer has the same periodicity quantum beat, I would suggest the authors look at the FC active vibrations in indole monomer, and see whether there are aromatic ring distortion modes that could be involved.

Please see our response to reviewer 1. Briefly, we more clearly describe that the oscillations are related to a difference in frequencies of vibrations. Our experiments unfortunately do not provide information on the vibrations that lead to the oscillations, but we now included several examples of ring breathing motions that differ by $\sim 20\text{ cm}^{-1}$ and could therefore also lead to the oscillations.

5. The kinetic scheme presented in equation (1) gives the false impression that the four processes occur in parallel; however, in the explanation, the authors clearly are assuming that these are sequential processes. Please clarify this in the text. For instance, how is indole⁺ produced within 1 ps if processes 1-4 are sequential?

We have updated (1) in the manuscript to clarify that these are sequential processes. We moreover adjusted the description of (1) in the manuscript (l. 188–190).

The observations could be described well by a five-level model, allowing to populate the levels sequentially:

6. The authors assume in their analysis that all indole⁺ signal comes from complexes that have dissociated after UV excitation. The alternative is that some of the indole⁺ signal arises from photofragmentation of the ionized complex. It seems unlikely to me that experimental conditions stopped absorption of the $1.36\text{ }\mu\text{m}$ laser pulse after a 3-photon process to form indole- H_2O^+ ,

while in indole⁺ the process involves at least 4 photons. Did the authors determine any power dependence to their ion signals?

No, we do not assume that all indole⁺ signal is due to dissociation, but indeed did also take into account dissociative ionisation throughout the time evolution. We referred to this process as 'fragmentation of indole(H₂O)⁺'. We have now updated the manuscript and Figure 3 to clarify this further.

Carefully determining the power dependence of the different ion signals as function of the pump-probe delay is not as straightforward as it might seem, due to the different pathways that are involved in the dynamics, and therefore beyond the scope of the current work. In test experiments, we tried to determine the power dependence of the ion signals, but these measurements were unfortunately inconclusive.

7. The weak time dependence to the H₂O⁺ signal – are you fitting with a zero offset of 0.08 ions/shot?

As written in the Methods section, we fitted simulated against background-corrected experimental ion signals. This background – or zero offset – was determined by taking the mean value of the ion signals for negative delays. For the H₂O⁺ signal, this resulted in an offset of 0.0081 ions/shot. Note that the H₂O⁺ signal in Figure 1a of the manuscript is magnified by a factor 10, as stated in the legend.

8. At times, it is hard to filter out what in their explanations comes from previous work and what they are deducing on their own. For instance, on p. 4 of the manuscript (upper right), the authors state that "the N-O distance decreases, whilst the N-H distance increases with respect to the geometry in the S₀ state." However, their experiment has no way of deducing this.

We have updated the manuscript to clearly state which details are from previous literature, the analysis of our experimental results, or accompanying calculations. Please see the PDF file with changes marked for the details.

A couple of minor points:

1. The authors should state explicitly whether there was any orientation dependence in the VMI images.

We did not observe any orientation dependence. In fact, we know from other experiments (Mullins *et al.*, 2022) that we do not expect any significant orientation to occur for the DC fields used in this experiment. After careful considerations, we refrain from stating this explicitly in the manuscript, since we fear that it would confuse the reader more than it would help.

2. The authors use sometimes misleading language, seemingly in order to make a connection to other issues. Two examples: (1) 'solvated electron' when in fact this same orbital shape is present in the absence of the H₂O molecule in the indole monomer. (2) 'bimolecular solute-solvent aggregate' rather than 'indole-H₂O complex'.

The phrase "solvated electron", or "electron solvation" in this context was brought up by Sobolewski and Domcke (2000, [32 in the manuscript]). We also referred to this reference in our manuscript where we mentioned "solvated electron".

Regarding the "bimolecular solute-solvent aggregate" we do not see any issue with this term, as the "indole-H₂O complex" in fact *is* a "bimolecular solute-solvent aggregate". And in connection to the radiation-damage aspects of our work, the fact that in biological systems water is abundant as solvent around indole-chromophore containing proteins renders this phrase useful in our opinion. Nevertheless, we have carefully considered our wording in light of this comment, please see the PDF file with changes marked for the details.

Reviewer 3

This manuscript describes a femtosecond time-resolved ion-yield measurement on an indole-water beam created by electrostatic deflection. The indole-water complex is excited by a 650-fs long ultraviolet pulse and its dynamics are probed by strong-field ionization with a 1320-nm laser pulse. The authors interpret the time-resolved ion yields in terms of a five-level kinetic model, which consists of photoexcitation to the $\pi\text{-}\pi^*$ state, followed by internal conversion to the $\pi\text{-}\sigma^*$ state, electronic relaxation to the S0 ground state and statistical dissociation on the ground state. In addition to the ion yields, the authors also measured and interpreted the kinetic-energy release (KER) of the H_2O^+ ions generated by the probe pulse.

The novelty of this work resides in the fact that excited-state dynamics are studied in a species prepared with the electrostatic-deflection technique. This is a very welcome and timely development, which opens many interesting possibilities. For this reason, I believe that this manuscript could eventually be published in Nature Communications.

Thank you for the detailed summary and positive evaluation.

However, although the experimental results are of high quality and the presentation of the data is compelling, I have some reservations with regards to the interpretation of the experimental data. I would suggest that the authors support their interpretation with more rigorous theoretical results, including dynamical calculations, which are nowadays well within reach using standard quantum-chemistry packages.

We appreciate the positive comment about our experiment and its results. Following the reviewers comments we have also extensively discussed the results with theoretical chemists, including experts at CFEL in Hamburg (Santra and Rubio groups) as well as with Oriol Vendrell in Heidelberg. We provide specific feedback below, but the overall summary is that actual quantitative calculations that provide any significant improvements in theoretical description over the current analysis would be very challenging, time consuming to the level that they would require an additional graduate student or postdoc in these groups, and thus well beyond the scope of the current manuscript. Nevertheless, we did clarify our discussions in line with this reviewers comment(s) and including details from the further discussions we had on theory. We believe that the interpretation is solid as it stands, but better clarified the rationale and origin of the different aspects of the interpretation.

Specifically, my recommendations are the following: 1) It is not clear how to reconcile the potential-energy surfaces shown in Fig. 3 with the ionization and dissociation energies given in the figure and the text. The horizontal line at 7.37 eV and the repulsive curve dissociating to $\text{indole}^+ + \text{H}_2\text{O}$ are both supposed to represent the electronic ground state of $\text{indole-H}_2\text{O}^+$. I suggest that the authors calculate the actual potential energy curve of this electronic ground state and display it in Fig. 3.

We do understand that Figure 3 is very rich in information and somewhat complex. Based on the reviewers comments and further discussions, we clarified Figure 3 and its description in the manuscript, please see the PDF file with changes marked for further details.

While we have performed quantum-chemistry calculations for the ground-states of the neutral molecules, the suggested open-shell calculations of this still relatively big system are beyond the scope of the manuscript – esp. considering the quite clear description available based on literature rationalisation, partly even experimental insight (Mons *et al.*, 1999, [46]).

2) As drawn, Fig. 3 suggests that $1\text{UV} + 3\text{IR}$ photons can reach 7.76 eV, but the sum of the photon energies is only 7.43 eV. Probably 4-photon processes are not much less likely than 3-photon processes at 10^{13} W/cm² and the authors actually observe 4-photon ionization?

We apologize for this inconsistency in our manuscript and thank the referee for correctly pointing this out. The referee is correct that this indeed is a 4-photon process, and that this process is

likely to occur given the intensities we used in the experiment. We updated this in the figure and the text of the manuscript. We moreover noticed that the figure indicated that 13 probe photons are needed to ionise neutral H₂O molecules, which should be 14 as was written in the text.

3) The authors assign $\tau_2=445$ fs to the $\pi\text{-}\pi^* \rightarrow \pi\text{-}\sigma^*$ electronic relaxation. This is unusually slow for this type of internal conversion. Moreover, this interpretation contrasts with Ref. 31, which assigned a ~ 50 -fs time scale to this process. Although Ref. 31 did not resolve the individual indole-water clusters, it is hard to imagine a factor 10 difference of this time scale as a function of cluster size. I suggest that the authors verify their interpretation by comparison with dynamical calculations.

We note that the $\tau_2 = 445$ fs timescale does not reflect solely the electronic transition, but it corresponds to the timescale to go from state 2 to state 3, as defined in the manuscript and observed in our experiment, by changing the ionisation products, i. e., the change from simple ionisation to dissociative ionisation and the production of indole⁺. This includes relevant nuclear rearrangements, of the neutral species, toward the $\pi\sigma^*$ equilibrium structure depicted in the inset of Figure 3. In that respect, this timescale seems completely reasonable to us.

In fact, Peralta Conde *et al.* (2012), [39]) attributed a 150-500 fs process to dynamics along the $\pi\sigma^*$ state, which is in agreement with our assignment.

Previous literature was also not decisive on the assignment of the different timescales. The fastest timescales of 20-100 fs (Peralta Conde *et al.*, 2012, [39]) or ~ 50 fs (Lippert *et al.*, 2003, [31]) were attributed to internal conversion between the L_a and L_b states or between the $\pi\pi^*$ and $\pi\sigma^*$ states, respectively. We unfortunately were not able to observe this fast timescale due to the relatively large instrument response function in our experiment.

4) Related to 3), it would be helpful to understand why the UV pulse was so long (650 fs), inspite of the 4.5 nm bandwidth. Was it stretched on purpose? If it was stretched, it must be highly chirped and the chirp might have an influence on the excited-state dynamics.

The long pulse duration was unintentional, due to experimental artifacts, and unfortunately only revealed after the experiment. As data analysis still yields a very clear picture of the ongoing dynamics and redoing the experiments with a 50–100 fs temporal resolution as possible in our experiment would be a major experimental campaign, we retreated from this. In fact, we are planning to perform the experiment at much higher temporal resolution of a few fs in collaboration with the Calegari group (Popova-Gorelova *et al.*, 2016; Galli *et al.*, 2019) to provide completely new insight into the ultrafast electronic processes of such complex systems – at an even significantly higher level of experimental complexity. Nevertheless, we are sure that the current analysis and thus the derived timescales are robust and demonstrate the capabilities of forefront experimental approaches for complex molecular systems.

5) The experimental work is of high quality and the study of excited-state dynamics in species separated by electrostatic deflection is innovative. I nevertheless believe that the title "solvated biomolecules" is an exaggeration and should be toned down to describe more accurately what is being reported.

Thank you for the positive evaluation of our work. In light of your criticism of the title we updated it.

6) The explanation that is given for the oscillation with a period of 1.67 ps is not clear. The authors argue that the 20 cm⁻¹ interval corresponds to the difference of the frequencies of C-H stretching vibrations. I am not convinced by this explanation because the absence of dephasing suggests that only two quantum states are prepared and it is hard to imagine such a selectivity among the C-H stretching vibrations of indole. I suggest that the authors carefully study other possible explanations for this observation.

We have carefully considered this question, the related questions by the other referees, and

corresponding literature, based on which we included an alternative explanation in the manuscript. Please see our response to reviewer 1 for more details.

Overall, I view the reported experimental work as a very important and promising development at the forefront of ultrafast molecular science. Unfortunately, I find the current interpretation is not sufficiently solid or convincing to recommend publication at this point.

We thank the reviewer for the generally very positive evaluation of our work and the support of it as very important progress in ultrafast molecular science. We have carefully considered all specific points brought up regarding the analysis and interpretation of the experimental results and believe that the results and interpretation are solid – and we hope that with our updated manuscript and the explanations above we can also convey this “robustness” to this reviewer.

References

- S. Trippel, Y.-P. Chang, S. Stern, T. Mullins, L. Holmegaard, and J. Küpper, Spatial separation of state- and size-selected neutral clusters, Phys. Rev. A **86**, 033202 (2012), arXiv:1208.4935 [physics].
- S. Trippel, M. Johny, T. Kierspel, J. Onvlee, H. Bieker, H. Ye, T. Mullins, L. Gumprecht, K. Długolecki, and J. Küpper, Knife edge skimming for improved separation of molecular species by the deflector, Rev. Sci. Instrum. **89**, 096110 (2018), arXiv:1802.04053 [physics].
- M. Mons, I. Dimicoli, B. Tardivel, F. Piuzzi, V. Brenner, and P. Millié, Site dependence of the binding energy of water to indole: Microscopic approach to the side chain hydration of tryptophan, J. Phys. Chem. A **103**, 9958–9965 (1999).
- A. L. Sobolewski and W. Domcke, Photoinduced charge separation in indole–water clusters, Chem. Phys. Lett. **329**, 130–137 (2000).
- H. Lippert, *Ultrakurzzeitspektroskopie von isolierten und mikrosolvatisierten Biochromophoren*, Ph. D. thesis, Freien Universität Berlin, Berlin, Germany (2004).
- A. Peralta Conde, V. Ovejas, R. Montero, F. Castaño, and A. Longarte, Influence of solvation on the indole photophysics: Ultrafast dynamics of indole–water clusters, Chem. Phys. Lett. **530**, 25–30 (2012).
- T. M. Korter, D. W. Pratt, and J. Küpper, Indole-H₂O in the gas phase. Structures, barriers to internal motion, and S₁ ← S₀ transition moment orientation. Solvent reorganization in the electronically excited state, J. Phys. Chem. A **102**, 7211–7216 (1998).
- T. Mullins, E. T. Karamatskos, J. Wiese, J. Onvlee, A. Rouzée, A. Yachmenev, S. Trippel, and J. Küpper, Picosecond pulse-shaping for strong three-dimensional field-free alignment of generic asymmetric-top molecules, Nat. Commun. **13**, 1431 (2022).
- H. Lippert, V. Stert, L. Hesse, C. P. Schulz, I. V. Hertel, and W. Radloff, Ultrafast photoinduced processes in indole-water clusters, Chem. Phys. Lett. **376**, 40–48 (2003).
- D. Popova-Gorelova, J. Küpper, and R. Santra, Imaging electron dynamics with time- and angle-resolved photoelectron spectroscopy, Phys. Rev. A **94**, 013412 (2016).
- M. Galli, V. Wanie, D. P. Lopes, E. P. Månsson, A. Trabattoni, L. Colaizzi, K. Saraswathula, A. Cartella, F. Frassetto, L. Poletto, F. Légaré, S. Stagira, M. Nisoli, R. Martínez Vázquez, R. Osellame, and F. Calegari, Generation of deep ultraviolet sub-2-fs pulses, Opt. Lett. **44**, 1308 (2019).

REVIEWERS' COMMENTS

Reviewer #2 (Remarks to the Author):

The authors have responded adequately to the scientific issues raised in my review. All three reviewers saw a range of issues associated with the interpretation of these rich, but complicated time domain signals. Nevertheless, the manuscript has been improved by the review process and the remaining issues are likely to be the subject of theoretical work seeking to reproduce the experimental observables. From this reviewer's perspective, the manuscript can be published in Nature Communications in this revised form.

Reviewer #3 (Remarks to the Author):

The authors have thoroughly revised their manuscript and have clarified many important questions regarding the experimental aspects. Unfortunately, they have not been able to provide theoretical support for the interpretation of the time-dependent signals that they have observed. As stated in my previous report, the most critical point is the assignment of $\tau_2=445$ fs to the transition from "state 2 to state 3", i.e. $\pi\text{-}\pi^* \rightarrow \pi\text{-}\sigma^*$. This assignment contrasts with previous work, in particular with Lippert et al., 2003 [31] who assigned a ~ 50 fs time scale to this process, but it agrees with the relaxation time of $\tau_2=385\pm 237$ fs observed by A. Peralta-Conde, 2012 (see their Fig. 5), following excitation at 260 nm. Since the interpretation of this latter work relies on accurate ab-initio calculations, I am willing to accept this interpretation.

Overall, I would thus recommend that the authors revise their manuscript by referring more explicitly and accurately to the previous literature, in particular to the fact that their experiments are expected to populate both the La and the Lb states (both $\pi\text{-}\pi^*$ in character), that the La state is thought to relax to Lb within 30-40 fs and that their interpretation of the time constants is consistent with Peralta-Conde et al. I also recommend that they add the calculated and experimental excitation energies of the La, Lb and $\pi\text{-}\sigma^*$ states in Fig. 3, such that these informations are readily available to the reader.

Response to reviewer comments

Reviewer 2

The authors have responded adequately to the scientific issues raised in my review. All three reviewers saw a range of issues associated with the interpretation of these rich, but complicated time domain signals. Nevertheless, the manuscript has been improved by the review process and the remaining issues are likely to be the subject of theoretical work seeking to reproduce the experimental observables. From this reviewer's perspective, the manuscript can be published in Nature Communications in this revised form.

Thank you very much for the appreciation of our revisions and response and for recommending our revised manuscript for publication in Nature Communications.

Reviewer 3

The authors have thoroughly revised their manuscript and have clarified many important questions regarding the experimental aspects.

Thank you for these positive words.

Unfortunately, they have not been able to provide theoretical support for the interpretation of the time-dependent signals that they have observed. As stated in my previous report, the most critical point is the assignment of $\tau_2=445$ fs to the transition from "state 2 to state 3", i.e. $\pi\pi^* \rightarrow \pi\sigma^*$. This assignment contrasts with previous work, in particular with Lippert et al., 2003 [31] who assigned a 50 fs time scale to this process, but it agrees with the relaxation time of $\tau_2=385\pm 237$ fs observed by A. Peralta-Conde, 2012 (see their Fig. 5), following excitation at 260 nm. Since the interpretation of this latter work relies on accurate ab-initio calculations, I am willing to accept this interpretation.

We would like to note that our transition from state 2 to state 3 does not correspond to a pure electronic transition from the $\pi\pi^*$ to the $\pi\sigma^*$ state, but also includes the accompanied nuclear rearrangement and electron transfer, as stated in the main manuscript (l. 237-239 and 371-375). This could explain why it is not in agreement with Lippert *et al.* (2003, [31 in the manuscript]), who assigned a 50 fs timescale to the pure electronic transition.

Overall, I would thus recommend that the authors revise their manuscript by referring more explicitly and accurately to the previous literature, in particular to the fact that their experiments are expected to populate both the La and the Lb states (both $\pi\pi^*$ in character), that the La state is thought to relax to Lb within 30-40 fs and that their interpretation of the time constants is consistent with Peralta-Conde et al.

We made some changes to the manuscript to incorporate these suggestions. In the introduction, we explicitly added the following information about the $\pi\pi^*$ states (l. 97-98):

..., in the case of indole most likely two, i. e., 1L_b and 1L_a [41].

We also explicitly state that our model does not consider them as individual states (l. 204-206)

We note ... and that the model does not distinguish between the 1L_b and 1L_a states, but rather considers them as one state.

In the discussion, we more explicitly referred to the literature (l. 377-383):

We could not resolve the fastest time constant of ~ 50 fs [31] or 20...100 fs [39] observed before, since it is shorter than the IRF. This time constant was previously assigned to internal conversion from the $\pi\pi^*$ to the $\pi\sigma^*$ state [31] or from the 1L_a to the 1L_b state [39], which are both $\pi\pi^*$ states.

We already stated that the value we determined for τ_2 is in agreement with the analysis by Peralta Conde *et al.* (2012), [39 in the manuscript]) (l. 371-375).

I also recommend that they add the calculated and experimental excitation energies of the La, Lb and pi-sigma* states in Fig. 3, such that these informations are readily available to the reader.

We refrained from including this information in Fig. 3, since we think that this renders the figure confusing: i) Our model does not consider the two $\pi\pi^*$ states separately, and ii) as the $\pi\sigma^*$ state is dissociative, there is not one specific energy that we can mention here.

References

- H. Lippert, V. Stert, L. Hesse, C. P. Schulz, I. V. Hertel, and W. Radloff, Ultrafast photoinduced processes in indole-water clusters, Chem. Phys. Lett. **376**, 40–48 (2003).
- A. Peralta Conde, V. Ovejas, R. Montero, F. Castaño, and A. Longarte, Influence of solvation on the indole photophysics: Ultrafast dynamics of indole–water clusters, Chem. Phys. Lett. **530**, 25–30 (2012).